# Adverse Effects of Intravesical OnabotulinumtoxinA Injection in Patients with Idiopathic Overactive Bladder or Neurogenic Detrusor Overactivity: A Systematic Review and Meta-Analysis of Randomized Controlled Studies

**DOI:** 10.3390/toxins16080343

**Published:** 2024-08-05

**Authors:** Ping-Hsuan Yu, Chung-Cheng Wang

**Affiliations:** 1Department of Urology, En Chu Kong Hospital, College of Medicine, National Taiwan University, New Taipei City 237414, Taiwan; benbenben642001@hotmail.com; 2Department of Urology, Taipei Veterans General Hospital, Taipei 112201, Taiwan; 3Department of Urology, College of Medicine, National Yang Ming Chiao Tung University, Taipei 112304, Taiwan; 4Shu-Tien Urological Science Research Center, National Yang Ming Chiao Tung University, Taipei 112304, Taiwan; 5Department of Biomedical Engineering, Chung Yuan Christian University, Taoyuan 320314, Taiwan

**Keywords:** adverse effects, neurogenic detrusor overactivity, idiopathic overactive bladder, onabotulinumtoxinA, meta-analysis

## Abstract

Despite the efficacy of onabotulinumtoxinA, its safety profile remains a concern. This meta-analysis reviewed the major adverse events (AEs) associated with intravesical onabotulinumtoxinA treatment in patients with neurogenic detrusor overactivity (NDO) and idiopathic overactive bladder (iOAB). Randomized controlled trials (RCTs) conducted between January 2000 and December 2022 were searched for adult patients administered different onabotulinumtoxinA dosages or onabotulinumtoxinA vs. placebo. Quality assessment was performed using the Cochrane Collaboration tool, and statistical analysis was performed using Review Manager version 5.3. A total of 26 RCTs were included in the analysis, including 8 on NDO and 18 on iOAB. OnabotulinumtoxinA vs. placebo significantly increased the urinary tract infection (UTI) incidence in patients with NDO (relative risk, or RR, 1.54) and iOAB (RR, 2.53). No difference in the RR with different onabotulinumtoxinA dosages was noted. Urinary retention was frequent with onabotulinumtoxinA use in the NDO (RR, 6.56) and iOAB (RR, 7.32) groups. Similar observations were made regarding the risks of de novo clean intermittent catheterization (CIC). The risk of voiding difficulty increased with onabotulinumtoxinA use in patients with iOAB. Systemic AEs of onabotulinumtoxinA, including muscle weakness (RR, 2.79) and nausea (RR, 3.15), were noted in patients with NDO; most systemic AEs had a low incidence and were sporadic.

## 1. Introduction

Overactive bladder (OAB) is defined as urinary urgency with or without urgency urinary incontinence, along with other storage symptoms such as daytime frequency and nocturia, without evidence of urinary tract infection (UTI) or other pathological entities [1,2]. Based on the existence of underlying neurological diseases, OAB can be further classified into two different types: neurogenic detrusor overactivity (NDO) and idiopathic overactive bladder (iOAB). NDO generally results from multiple sclerosis (MS) or spinal cord injury (SCI) [3,4]. The prevalence of OAB ranges from 12% to 19%, and it is a common health problem that has a negative impact on many patients physically, mentally, and socially [3,5,6].

Multiple approaches have been applied for the treatment of OAB, ranging from non-pharmacological treatments, such as behavioral modifications and pelvic floor muscle training, to medications, including antimuscarinics and beta-3 adrenoreceptor agonists [7,8]. However, some patients still exhibit refractory symptoms with oral medications, while others may be intolerant to a high rate of adverse effects (AEs), including xerostomia, blurred vision, constipation, cognitive dysfunction, or high blood pressure [9,10]. In such circumstances, intravesical onabotulinumtoxinA injection, posterior tibial nerve stimulation, and sacral neuromodulation are invasive but feasible alternative treatment options [11,12].

OnabotulinumtoxinA, a specific formulation of botulinum toxin A, is a neurotoxin produced by the anaerobic bacterium *Clostridium botulinum* [13]. It inhibits acetylcholine release from presynaptic neurons by binding to synaptic vesicle glycoprotein 2A, thereby paralyzing muscle contraction [14,15]. The use of intravesical onabotulinumtoxinA injection was first reported by Schurch in 2000 in patients with NDO to reduce excessive detrusor spasticity [16]. After injection with onabotulinumtoxinA, the treatment effects can last for at least 6 months [17]. Gradually, onabotulinumtoxinA treatment has been widely accepted for patients with NDO and iOAB. Intravesical onabotulinumtoxinA injections were approved by the Food and Drug Administration in 2011 to treat urinary incontinence associated with NDO. Currently, the American Urological Association and European Association of Urology guidelines recommend intravesical onabotulinumtoxinA injection as a third-line treatment for OAB [18,19].

Although its efficacy and duration of effectiveness are well recognized, the safety profile of onabotulinumtoxinA remains a concern. Intravesical onabotulinumtoxinA injection causes AEs, particularly a substantially high incidence of UTI and urinary retention [20,21]. Injection of onabotulinumtoxinA inevitably increases the post-void residual volume, thereby increasing the risk of urine retention, de novo clean intermittent catheterization (CIC), and UTI [22]. In addition to AEs localized to the urinary tract, other systemic AEs such as muscle weakness, fatigue, and autonomic dysreflexia have been occasionally reported [23,24].

In this study, we primarily focused on the major AEs associated with intravesical onabotulinumtoxinA treatment, including localized and systemic AEs, in patients with NDO or iOAB. The incidence of each AE under onabotulinumtoxinA injection was recorded, and the increasing risk between different dosages and placebo treatment was evaluated using a meta-analysis.

## 2. Results

### 2.1. Study Selection, Characteristics, and Risk of Bias

This review was performed in accordance with the Preferred Reporting Items for Systematic Reviews and Meta-Analyses (PRISMA) guidelines. Initially, 1526 literature sources were identified, and 260 were eliminated owing to duplication. After screening the titles and abstracts, 1185 articles were excluded, leaving only 81 articles for full-text reading. Finally, 26 articles were included in this meta-analysis (Figure 1) [23,24,25,26,27,28,29,30,31,32,33,34,35,36,37,38,39,40,41,42,43,44,45,46,47,48].

This review encompassed 26 literature sources pertaining to randomized controlled trials (RCTs) on bladder onabotulinumtoxinA injections (Table 1). These studies involved 3876 patients diagnosed with NDO or iOAB. Eighteen RCTs, comprising 2904 patients, were related to iOAB (1113 in the placebo group and 1640 in the onabotulinumtoxinA group, with the 151 in the solifenacin group not being analyzed), whereas eight RCTs, including 972 patients, were related to NDO (379 in the placebo group and 593 in the onabotulinumtoxinA group). The etiologies of NDO included MS (*n* = 550) and SCI (*n* = 422).

In the OAB literature, the studies by Dmochowski et al. [35] and Rovner et al. [38] were related to the same cohort in a phase II dose-ranging trial. Since most AEs were detailed in the study by Dmochowski et al., we listed all the literature in Table 1 but only performed an analysis of the study by Dmochowski et al. In addition, Chapple et al. [41] and Nitti et al. [42] studied two cohorts from the EMBARK study group, and Sievert et al. [44] presented the combined data from these two cohorts. We listed the three literature sources; however, most of the analyses focused on the first two articles. Similar conditions existed in the NDO literature, with the studies by Cruz et al. [23], Ginsberg et al. [24], Ginsberg et al. [27], and Rovner et al. [28] all related to the DIGNITY study. This analysis was primarily carried out on the studies by Cruz et al. [23] and Ginsberg et al. [24]. The OAB study by Herschorn et al. [46] included onabotulinumtoxinA, solifenacin, and placebo treatment arms. Our analysis compared the onabotulinumtoxinA arm with a placebo.

Regarding the dose of onabotulinumtoxinA, the onabotulinumtoxinA units in the iOAB trials ranged from 50 U to 300 U, with 100 U being the most commonly administered dose, followed by 200 U. In the NDO trials, most doses were between 200 U and 300 U, except for one RCT in which 100 U botulinum toxin was injected into patients with MS [29]. The AEs have been monitored at different time points in various studies. The incidence of AEs analyzed in this study was mainly focused on the first 24 weeks after the injection.

The risk of bias in all the studies was assessed by two reviewers using the Cochrane Collaboration tool. Table 2 presents the results of the studies.

### 2.2. Adverse Effects Localized to Urinary Tract

In all the articles, most AEs were localized to the urinary tract, with minimal or no systemic effects. In addition, most AEs were transient and reversible. UTI, urinary retention, and voiding difficulties were the most prevalent AEs. 

#### 2.2.1. Urinary Tract Infections

UTI was the most frequently mentioned AE in all the studies. The definition of UTI varied among the studies. The most common definition was bacteriuria, along with more than five white blood cells in the high-power field. Some studies defined UTI based solely on urine strip tests or urine cultures, whereas others did not provide a clear definition. 

All eight articles involving patients with NDO reported the incidence of UTI. At baseline, the pooled UTI rate in the NDO placebo group was higher than that in the iOAB placebo group (29.2% vs. 9.4%). With increased treatment dosages for NDO, the pooled incidence of UTI in the onabotulinumtoxinA group was 48.4% (relative risk [RR], 1.54; confidence interval [CI], 1.30–1.83; *p* < 0.00001) (Figure 2). The RRs for the 200 U vs. placebo and 300 U vs. placebo groups were similar (1.47 and 1.46). Consequently, no significant difference in the incidence of UTI was observed between the 300 U and 200 U groups (RR, 1.06; CI, 0.89–1.26; *p* = 0.51) (Appendix A).

When comparing the two etiologies of NDO, most patients with SCI had already regularly undergone CIC. Therefore, we analyzed the incidence of UTI among the two etiologies separately. The RR of the incidence of UTI in patients with MS treated with onabotulinumtoxinA was 2.08 (CI, 1.58–2.75; *p* < 0.00001), which was higher than the RR of all the patients with NDO and patients with SCI (Figure 2). The incidence of UTI between the 300 U and 200 U dosages in patients with MS showed no significant difference (RR, 1.10; CI, 0.88–1.37; *p* = 0.40) (Appendix A). On the contrary, in patients with SCI, no difference in risk between the onabotulinumtoxinA and placebo groups was observed (RR, 1.17; CI, 0.91–1.50; *p* = 0.23) (Figure 2).

Regarding the patients with iOAB, we analyzed 14 studies that evaluated the UTI incidence following onabotulinumtoxinA treatment. The incidence of UTI between the combined onabotulinumtoxinA and placebo groups showed a significant difference (RR, 2.53; CI, 2.05–3.11; *p* < 0.00001) (Figure 2). The incidence of UTI in the onabotulinumtoxinA 100 U group was 22.7%, compared with 9.4% in the placebo group. The incidence of UTI in the onabotulinumtoxinA 200 U group was 32.1%. Although a dose-dependent rising trend was observed, the RR when comparing 200 U to 100 U did not reach a significant difference (RR, 1.44; CI, 0.94–2.20; *p* = 0.09) (Appendix A).

Acute pyelonephritis (APN) is a relatively severe AE that can develop even after treatment. APN was mentioned in two studies on iOAB and three studies on NDO, and it appeared to be more sporadic than drug-related APN. One patient in the onabotulinumtoxinA 100 U group and another patient in the 50 U group experienced APN after treatment for iOAB. Three patients in the placebo group for NDO and one patient in the 300 U group were diagnosed with APN after treatment. 

Urosepsis is worth noting because of its potentially lethal nature. This issue was addressed in two main studies on NDO [23,24], and it appeared to be more related to the invasive procedure than to onabotulinumtoxinA. Two patients in the placebo group experienced urosepsis, with an incidence of 0.9%.

#### 2.2.2. Urinary Retention

Urinary retention was another major AE commonly reported after treatment with onabotulinumtoxinA. In some studies, urinary retention was defined as a post-void residual volume of >200 mL requiring CIC. However, some studies have defined urinary retention based solely on clinical judgment. The proportion of male patients in each study may have led to distinct results.

Regarding NDO, two articles related to the DIGNITY study [23,24], Tullman et al. [29] and Honda et al. [30], investigated the incidence of urinary retention. The incidence of urinary retention was 20.4% in the onabotulinumtoxinA group, compared with 2.8% in the placebo group (RR, 6.56; CI, 3.43–12.54; *p* < 0.00001) (Figure 3). It is likely that many patients with SCI already underwent CIC (84.8% of patients with SCI underwent CIC at baseline, according to the data of the DIGNITY study [23,24,27,28]). Therefore, the evaluation of urinary retention in this patient group was not straightforward. 

By focusing only on patients with MS, the RR could reach 7.82 (CI, 3.71–16.49; *p* < 0.00001) (Figure 3). The combined incidence of urinary retention in patients with SCI was 1.7%, 7.4%, and 4.0% in the placebo, 200 U, and 300 U groups, respectively, whereas the combined incidence of urinary retention in patients with MS was 3.4%, 15.2%, 29.5%, and 39.3% in the placebo, 100 U, 200 U, and 300 U groups, respectively. Although a dose-dependent increasing trend in risk was observed, there was no significant difference between the 200 U and 300 U treatments in all the patients with NDO or the MS-only patients (Appendix A).

Although most patients with SCI already undergo regular CIC, the incidence of de novo CIC in patients with SCI who are yet to undergo CIC and in patients with MS remains a concern in terms of higher onabotulinumtoxinA injection doses. For all the patients with NDO without a history of CIC, the risk of de novo CIC for all causes was 2.61 times higher than that for the placebo group; for the MS-only patients, the risk was 2.92 times higher than that for the placebo group (Figure 3). However, the difference in the treatment dose did not affect the risk in all the patients with NDO or the patients with only MS (Appendix A).

Regarding iOAB, ten studies addressed urinary retention. Accordingly, the pooled retention rates were approximately 1.0% in the placebo group, 6.7% in the onabotulinumtoxinA 100 U group, and 20.1% in the onabotulinumtoxinA 200 U group. The risk of urinary retention for all the onabotulinumtoxinA doses was 7.32 times higher than that for the placebo group (CI, 3.95–13.58; *p* < 0.00001) (Figure 4). When comparing 200 U to 100 U, the increased risk of urinary retention was not significant (RR, 1.34; CI, 0.66–2.72; *p* = 0.42) (Appendix A). Several studies including patients with iOAB have objectively evaluated the volume of residual urine before and after treatment. Using a cutoff value of 200 mL for residual urine, only 0.3% of patients in the placebo group had residual urine exceeding 200 mL after treatment. After treatment with onabotulinumtoxinA, the incidence of residual urine exceeding 200 mL was 9.6% (10.09 times the risk; CI, 3.80–26.81; *p* < 0.00001).

We also evaluated the possibility of de novo CIC in patients with iOAB. The combined de novo CIC rate in the placebo group was 1.0%. With 100 U onabotulinumtoxinA treatment, the incidence of de novo CIC increased to 6.8% and further increased to 21.0% with 200 U onabotulinumtoxinA treatment. The risk of de novo CIC with onabotulinumtoxinA treatment was 6.63 times higher than that with placebo (CI, 3.80–11.57; *p* < 0.00001) (Figure 4). Dmochowski et al. [35] compared the incidence of de novo CIC between the 200 U onabotulinumtoxinA and 100 U onabotulinumtoxinA groups. The RR was 1.94 with a CI of 0.77–4.86, which still lacked significance (Appendix A).

#### 2.2.3. Voiding Difficulty

Although not as severe as urinary retention, difficulty in voiding caused inconvenience to patients undergoing onabotulinumtoxinA treatment. Four articles on NDO reported the incidence of voiding difficulty after onabotulinumtoxinA injection. The risk of voiding difficulty with onabotulinumtoxinA injection did not significantly increase, possibly because many patients underwent regular CIC (Figure 5). Eight iOAB studies addressed this issue. The combined incidence of voiding difficulties was approximately 5.3% in the placebo group, including patients with iOAB. The incidence increased to 8.8% in the 100 U group and 21.9% in the 200 U group. We observed not only an increased risk within the combined onabotulinumtoxinA group compared with the placebo group (RR, 1.82; CI, 1.32–2.52; *p* = 0.0003) but also within the 200 U group compared with the 100 U group (RR, 2.25; CI, 1.08–4.71; *p* = 0.03) (Figure 5 and Appendix A).

#### 2.2.4. Hematuria

Hematuria was another potential AE. Generally, the risk of hematuria did not increase with onabotulinumtoxinA treatment in patients with NDO or iOAB (Appendix A). Eight articles on iOAB and six articles on NDO discussed this AE. In the iOAB studies, the combined incidence of hematuria was approximately 3.4% in the placebo group and 4.3% in the 100 U group. In the patients with NDO, the combined incidence of hematuria was approximately 4.3% for the placebo group, 5.5% for the 200 U group, and 6.8% in the 300 U group.

#### 2.2.5. Bladder Pain

Despite the fact that an increasing trend of bladder pain was observed with onabotulinumtoxinA injection in patients with NDO, the incidence of bladder pain in the onabotulinumtoxinA group was not significantly higher than that in the placebo group (RR, 2.72; CI, 0.94–7.87; *p* = 0.07) (Appendix A). Four NDO articles reported the incidence of bladder pain. The pooled incidences were 0.9% in the placebo group, 1.2% in the 200 U group, and 5.1% in the 300 U group.

### 2.3. Systemic Adverse Effects 

The incidence of systemic AEs was generally low and these AEs usually manifested sporadically. The following were some of the most prevalent systemic AEs.

#### 2.3.1. Muscle Weakness 

Because of the muscle-relaxing effects of onabotulinumtoxinA, there are concerns about its systemic effects and the potential for muscle weakness in the extremities. Four studies on NDO and two on iOAB addressed this AE. For the patients with NDO, the combined incidences of muscle weakness were 1.8% in the placebo group, 4.6% in the 200 U group, and 6.6% in the 300 U group. The risk in the onabotulinumtoxinA group was 2.79 times higher than that in the placebo group (CI, 1.18–6.62; *p* = 0.02) (Figure 6). In the iOAB studies, the incidence of muscle weakness was generally low across the different treatment dosages. It was 0.8% in the placebo group and 0.7% in the 200 U group (Figure 6).

#### 2.3.2. Fatigue 

The articles from the DIGNITY study [23,24,27,28] evaluated the fatigue rate after injection in patients with NDO. The incidence of fatigue was higher in the patients with MS than in those with SCI who were treated with onabotulinumtoxinA (200 U: 11.6% vs. 1.0%, *p* = 0.001; 300 U: 6.0% vs. 0%, *p* = 0.016). The combined incidences of fatigue were 3.0% in the placebo group, 7.1% in the 200 U group, and 3.2% in the 300 U group. No difference in risk was observed between the onabotulinumtoxinA and placebo groups (Appendix A).

#### 2.3.3. Symptoms Related to the Digestive System (Nausea, Diarrhea, and Constipation)

AEs related to digestive symptoms, including nausea, diarrhea, and constipation, were reported in numerous NDO studies. The risk of diarrhea and constipation did not increase with onabotulinumtoxinA injection (diarrhea: RR, 1.11; CI, 0.59–2.09; *p* = 0.75, constipation: RR, 1.67; CI, 0.72–3.86; *p* = 0.23). However, the risk of nausea significantly increased with onabotulinumtoxinA injection (RR, 3.15; CI, 1.27–7.81; *p* = 0.01). The combined incidences of nausea were 1.9% in the placebo group, 4.0% in the 200 U group, and 7.0% in the 300 U group (Appendix A).

#### 2.3.4. Pyrexia

The articles from the DIGNITY study [23,24], Schurch et al. [25], and Honda et al. [30] evaluated the incidence of pyrexia. Among the patients with NDO, the pooled risk of pyrexia after treatment was 3.0% in the placebo group, 6.6% in the 200 U group, and 2.6% in the 300 U group. This incidence appeared to be unrelated to the treatment dosage (Appendix A).

#### 2.3.5. Autonomic Dysreflexia

With an increasing possibility of urinary retention, the risk of autonomic dysreflexia may also increase after onabotulinumtoxinA treatment, especially in patients with SCI. Four NDO articles were used to evaluate the risk. The pooled incidence of autonomic dysreflexia was 0.4% in the placebo group. When treated with 200 U onabotulinumtoxinA, the risk increased to 2.0%. In the 300 U group, the risk was 1.7%. The RR of autonomic dysreflexia did not increase significantly when comparing onabotulinumtoxinA treatment with placebo (Appendix A).

## 3. Discussion

Patients with NDO and iOAB have frequent urinary incontinence and other storage symptoms, which significantly reduce their quality of life [8,9,22]. Behavioral therapy, biofeedback, pharmacotherapy, electrical stimulation, onabotulinumtoxinA injection, and surgical intervention have been considered as effective modalities [7,8,49]. The pharmacotherapeutic agents for NDO and iOAB include muscarinic receptor antagonists and beta-3 agonists such as solifenacin, tolterodine, oxybutynin, and mirabegron. Nevertheless, a certain proportion of patients may complain of suboptimal efficacy or may not tolerate the related AEs [50,51,52]. Thus, researchers continue to search for other therapeutic options that provide long-term treatment efficacy with few AEs. In recent decades, intravesical onabotulinumtoxinA injections have been widely administered to patients with NDO and iOAB. While the application of onabotulinumtoxinA injection improves urinary symptoms and quality of life, onabotulinumtoxinA injection itself has corresponding AEs [20,53].

In this meta-analysis, we focused on the AEs related to intravesical onabotulinumtoxinA injections in patients with either iOAB or NDO in various RCTs. The AEs were mainly limited to the urinary system and were well tolerated. In the urinary system, the incidence of UTI, urinary retention, and de novo CIC induced by onabotulinumtoxinA treatment was higher than that by placebo in the patients with NDO and iOAB; however, the risk of each AE did not vary with an increase in dosage. An increasing incidence of voiding difficulty was observed solely in the patients with iOAB and tended to increase with increasing dosages. The risks of hematuria and injection pain were unrelated to onabotulinumtoxinA. Regarding the systemic AEs, increased risks of muscle weakness and nausea were observed in the patients with NDOs. Other systemic safety parameters, including autonomic dysreflexia, showed no clinically relevant changes.

Because intravesical onabotulinumtoxinA injection inhibits excessive neural signals, it reduces the strength of the detrusor muscle [14,15]. Although this decreases the symptoms of OAB, it results in incomplete bladder emptying, leading to urinary retention and sometimes necessitating catheterization for urine drainage. In the patients with either NDO or iOAB, an increase in urinary retention was observed following onabotulinumtoxinA treatment, along with a high proportion of patients who had never undergone CIC previously requiring CIC.

A phase II study involving patients with iOAB investigated multiple doses of onabotulinumtoxinA injections, including 50 U, 100 U, 150 U, 200 U, and 250 U [35,38]. It was found that any dose of onabotulinumtoxinA exceeding 100 U demonstrated a higher risk of urinary retention than that with the placebo. However, no statistically significant difference in the risk was observed among the higher doses. Surpassing a certain dosage threshold increases the likelihood of urinary retention; however, increasing the dose does not further increase the risk. Our meta-analysis, which compared the commonly used doses of 100 U and 200 U across multiple studies, yielded similar results.

Regarding NDO, the patients with MS and SCI comprised two distinct groups. According to the DIGNITY study, 29.4% of patients with MS had already undergone CIC before onabotulinumtoxinA treatment, whereas the proportion of patients with SCI was as high as 84.8%, making it worthwhile to investigate the two groups separately [23,24,27,28]. After onabotulinumtoxinA treatment, the patients with MS, many of whom lacked CIC, showed a significant increase in the risk of urinary retention and de novo CIC, similar to that in patients with iOAB. However, in the patients with SCI, the majority had already undergone CIC at baseline, and no increase in the related risk of CIC was observed.

Customarily, patients with MS, similar to patients with SCI, are treated with 200 U or 300 U of onabotulinumtoxinA to improve urinary incontinence. Tullman et al. treated patients with only 100 U of onabotulinumtoxinA [29]. When observing the incidence of urinary retention alone, the patients treated with 100 U of onabotulinumtoxinA had a slightly lower incidence than that of the patients treated with 200 U or 300 U of onabotulinumtoxinA. However, when compared with patients receiving placebo, the increased RR of urinary retention with 100 U was not less than that observed with 200 U or 300 U.

UTI was the most common AE after intravesical onabotulinumtoxinA injection. The incidence of UTI correlated with whether patients with NDO or iOAB experienced urinary retention after treatment. Among the patients with NDO, those with SCI underwent CIC at baseline at an increased proportion, which differentiated their RRs of UTI from those of the patients with MS. The DIGNITY study showed significantly different incidences of UTI with onabotulinumtoxinA vs. placebo in patients with MS compared to patients with SCI (MS: 56.1% vs. 29.2%; SCI: 51.0% vs. 44.8%) [23,24,27,28]. Notably, the increased incidence of UTI was more related to the increased post-void residual volume and urine retention than to the treatment dose. Therefore, the different treatment doses did not show a difference in the incidence of UTI in this analysis.

Although urinary retention increased the risk of UTI, intravesical onabotulinumtoxinA injection might have protected patients with OAB from developing complicated UTIs. One reason for this was that onabotulinumtoxinA treatment reduced the maximum detrusor pressure, thereby decreasing the occurrence of vesicoureteral reflux, which protected the kidneys from pyelonephritis [54]. Giannantoni et al. observed that after 6 years of onabotulinumtoxinA treatment, patients showed significant improvement in vesicoureteral reflux and renal pelvic dilatation [54]. Another reason was that lower bladder pressure could enhance the bladder blood flow and tissue oxygenation, which might prevent UTIs [55]. Overall, UTIs were primarily confined to the lower urinary tract, which is consistent with the findings of other meta-analyses [56,57,58].

However, some RCTs did not clearly address whether patients already had UTIs or whether bacteria were cultured from their urine before onabotulinumtoxinA injection. This may distort the evaluation of the incidence of UTI after onabotulinumtoxinA injection. Additionally, different studies had varying protocols regarding whether to maintain the use of antimuscarinic agents before and after onabotulinumtoxinA injection, which could also affect the incidence of UTIs. According to our meta-analysis, the patients with NDO treated with onabotulinumtoxinA or placebo generally had a higher UTI risk than that of the patients with iOAB. Therefore, it is recommended that patients with NDOs receive prophylactic antibiotics before injection to prevent post-treatment UTIs. 

Next, we discuss the potential systemic AEs associated with onabotulinumtoxinA injection. The patients with NDO generally required higher treatment doses, leading to greater concerns regarding systemic AEs. Most studies describing these systemic AEs were based on NDO. In our meta-analysis, the patients with NDO showed an increased risk of muscle weakness after intravesical onabotulinumtoxinA injection, whereas the patients with iOAB did not exhibit this phenomenon.

The effects of onabotulinumtoxinA spread from the primary injection site to the distal organs and produce related symptoms. Using onabotulinumtoxinA to treat muscle spasticity in children with cerebral palsy might carry a highest risk of muscle weakness, and symptoms could also occur in adults for other indications, such as NDO [21]. In addition to the potentially difficult patient transfer caused by muscle weakness, difficulties in swallowing or breathing can be life-threatening [21]. Nevertheless, when evaluating muscle weakness, other potential reasons for weakness should also be considered, including MS exacerbation, syringomyelia in SCI, and new-onset cerebrovascular accidents.

Nuanthaisong et al. retrospectively reviewed 13 patients with neurogenic bladder injected with cumulative doses of >360 U of onabotulinumtoxinA within a 3-month interval [59]. Four patients experienced general or extremity weakness without any life-threatening AEs, which were eventually resolved. In our meta-analysis, the doses of choice were 200 U and 300 U for the patients with NDO. The incidence of muscle weakness was low in general compared with that reported by Nuanthaisong et al. [59].

Autonomic dysreflexia may occur in patients with SCI with injuries at T6 or above. Noxious stimuli, such as instrumentation, bladder distention, or the injection itself, can induce autonomic dysreflexia. During the procedure, monitoring of the blood pressure and other vital signs is necessary for patients with high-level SCI, as “silent autonomic dysreflexia” may develop in approximately 40% of these patients [60,61]. However, in the long term, intravesical onabotulinumtoxinA injection had a positive effect on autonomic dysreflexia. Schurch et al. reported that three patients with tetraplegia and severe autonomic dysreflexia benefited from the resolution of autonomic dysreflexia with 300 U onabotulinumtoxinA injection [62].

This meta-analysis had several advantages. First, it included some recent studies published after 2015 in the analysis. This not only increased the number of patients analyzed but also ensured that the data reflected contemporary practice. Second, we systematically evaluated the literature for both NDO and iOAB. In addition to comparing the AEs between the treatment and control groups within each population, it allowed us to cross-examine the incidence of AEs between the NDO and iOAB groups. Finally, we compared whether different doses within the treatment group resulted in varying rates of AEs.

However, this study had some limitations. First, some of the included RCTs might have been terminated early owing to slow or difficult recruitment, resulting in a small sample size and high risk of bias. Second, only a few RCTs on NDO have assessed the safety profile of onabotulinumtoxinA injection. One reason for this is that the population of patients with NDO is inherently small, making it challenging to conduct related RCT studies. Additionally, the proportion of male and female patients in each study could affect the baseline status of whether patients have benign prostatic hyperplasia and further bladder outlet obstruction, which might influence the severity of the storage symptoms [63]. However, these RCTs did not separately report the incidence of AEs according to sex; thus, it was impossible to perform further subgroup analyses.

The heterogeneity in the reporting and definition of each AE is worth noting. The definition of a specific AE may vary across studies, and some studies may not provide a clear definition. For instance, in the case of UTI, the EMBARK study [41,42,44], which had the largest number of patients, and some other studies defined UTI as bacteriuria combined with more than five white blood cells per high-power field. However, some studies defined UTI based on patients’ self-reported symptoms [40], whereas others did not provide a precise definition. Finally, our meta-analysis focused on short-term AEs after intravesical onabotulinumtoxinA injection. Although repeated injections are pivotal for maintaining persistent therapeutic results, the effects of repeated injections or potential long-term AEs over a period longer than 1 year are beyond the scope of this meta-analysis. In view of this, as the articles by Chen et al. and Kennelly et al. primarily discussed repeated onabotulinumtoxinA treatments, they were not included in the analysis [64,65].

## 4. Conclusions

The risk of UTI, urinary retention, and de novo CIC increased after intravesical onabotulinumtoxinA injection in patients with either iOAB or NDO. The risks did not vary with increasing onabotulinumtoxinA doses. Most AEs after intravesical onabotulinumtoxinA injection were localized to the urinary tract and were controlled well.

## 5. Materials and Methods

This systematic review and meta-analysis was conducted according to the Preferred Reporting Items for Systematic Reviews and Meta-Analyses (PRISMA) statement [66].

### 5.1. Search Strategy

The PubMed, Ovid MEDLINE, Ovid Embase, and Cochrane Central Register of Controlled Trials databases were searched for relevant articles from 1 January 2000 to 31 December 2022 to evaluate the safety profiles of suburothelial or intradetrusor botulinum toxin injection treatment. RCTs related to onabotulinumtoxinA treatment in patients with NDO and iOAB were reviewed. The references listed in the retrieved studies were also searched. The search keywords included botulinum, botulinum toxin, Botox, botulinum neurotoxin A, onabotulinumtoxinA, neurogenic bladder, neurogenic detrusor overactivity, overactive bladder, and idiopathic overactive bladder.

### 5.2. Inclusion and Exclusion Criteria

The following criteria were used to include studies in the meta-analysis: (1) patients diagnosed with NDO (including SCI and MS) or iOAB who were refractory to oral antimuscarinics or beta-3 agonists or who were intolerant to the AEs; (2) adult patients aged >18 years; (3) randomized controlled design comparing onabotulinumtoxinA with placebo or onabotulinumtoxinA at different dosages; and (4) outcomes including AEs after the intervention, such as UTI, urinary retention, and hematuria. 

The exclusion criteria were as follows: (1) non-study participants, (2) single-arm studies, (3) studies not using onabotulinumtoxinA (e.g., abobotulinumtoxinA, rimabotulinumtoxinB, or incobotulinumtoxinA), (4) studies not evaluating AEs, and (5) studies focusing on repeated treatment instead of primary treatment. If multiple studies were related to the same cohort, all the studies presenting AEs were listed; however, the same patient group was analyzed only once.

### 5.3. Quality Assessment

To assess the methodological quality of the included RCTs, two reviewers (P.-H.Y. and C.-C.W.) employed the Cochrane Collaboration tool. The following seven domains were evaluated: blinding of participants and personnel, sequence generation, allocation concealment, blinding of outcome assessment, selective outcome reporting, incomplete outcome data, and other potential sources of bias. Each item was graded as “high risk”, “low risk”, or “unclear”.

### 5.4. Data Extraction

After the two reviewers reviewed the studies, the following details were collected from each article: first author, region, year of publication, population size, sex distribution, mean age, background diseases, study design, onabotulinumtoxinA treatment dosage, and follow-up intervals. The incidence of various AEs was recorded, including localized symptoms, such as UTI, urinary retention, incidence of CIC, voiding difficulty, and hematuria. Other systemic symptoms, such as muscle weakness and autonomic dysreflexia, were also noted. This study emphasized the short-term AEs following injection. In most studies, AEs were primarily observed within 6 months of injection. If the AEs were reported at multiple time points, priority was assigned to those that occurred within the first 12 weeks of treatment.

### 5.5. Statistical Analysis

The statistical analyses were conducted using Review Manager v. 5.3 (Cochrane Collaboration, Oxford, UK). Because the incidence of AEs was assessed as a dichotomous parameter, the data were expressed as RRs with 95% CIs [67,68]. A significance level of *p* < 0.05 was set for statistical significance. Forest plots were used to illustrate the outcomes. The I^2^ statistic provided an estimate of the percentage of heterogeneity, possibly due to chance [69], with the significance level set at *p* < 0.1. A fixed-effects model was used if the heterogeneity was not significant (I^2^ < 50%). In contrast, a random-effects model was used for the meta-analysis when heterogeneity was detected.

## Figures and Tables

**Figure 1 toxins-16-00343-f001:**
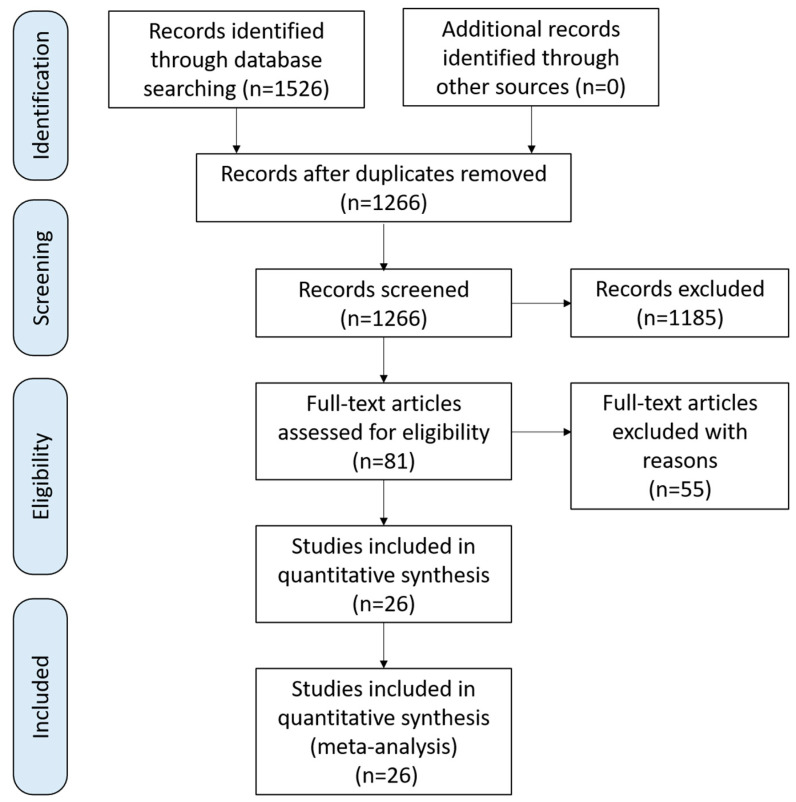
Flow diagram of the study participant selection process.

**Figure 2 toxins-16-00343-f002:**
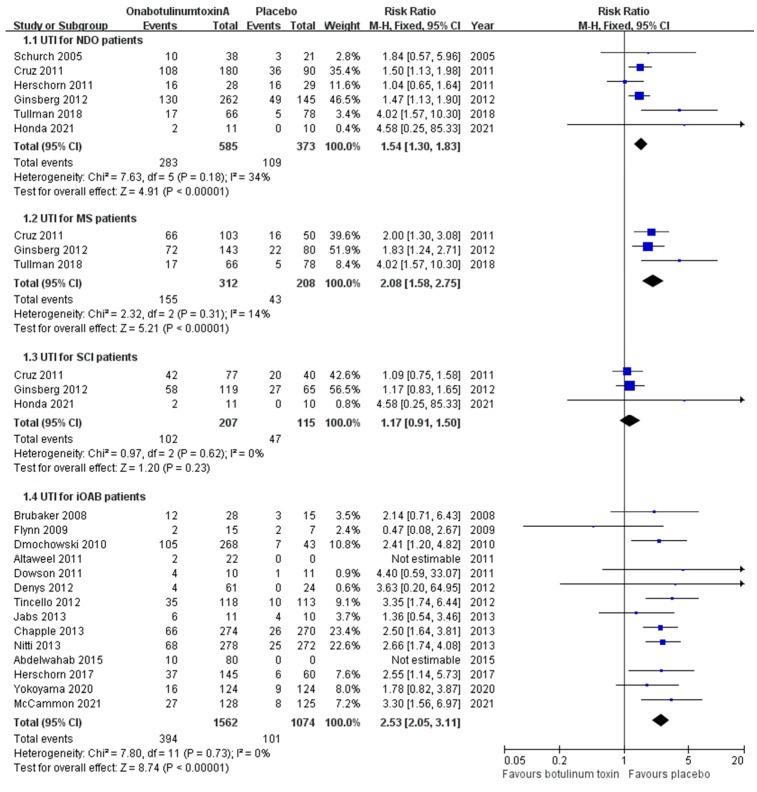
Forest plots of the incidence of urinary tract infection in patients with neurogenic detrusor overactivity due to all causes, multiple sclerosis, spinal cord injury, and idiopathic overactive bladder. iOAB: idiopathic overactive bladder; MS: multiple sclerosis; NDO: neurogenic detrusor overactivity; SCI: spinal cord injury; UTI: urinary tract infection.

**Figure 3 toxins-16-00343-f003:**
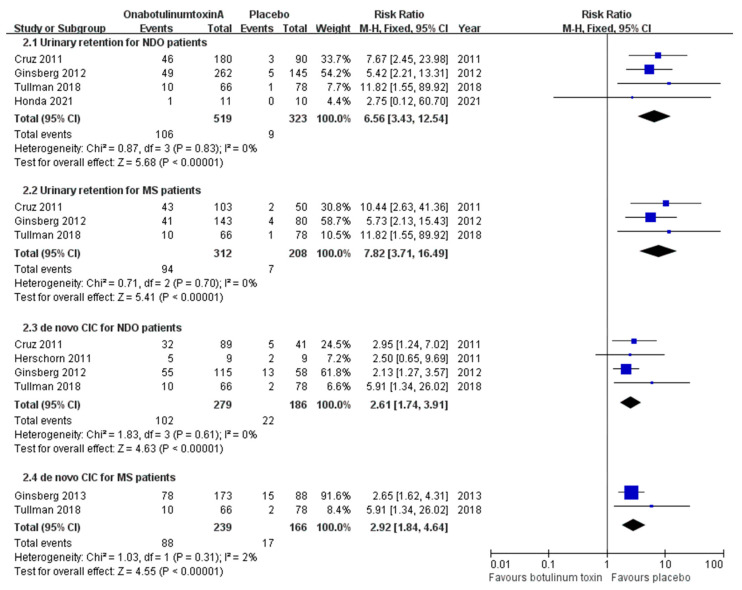
Forest plots of the incidence of urinary retention and de novo clean intermittent catheterization in patients with neurogenic detrusor overactivity due to all causes and neurogenic detrusor overactivity due to multiple sclerosis. CIC: clean intermittent catheterization; MS: multiple sclerosis; NDO: neurogenic detrusor overactivity.

**Figure 4 toxins-16-00343-f004:**
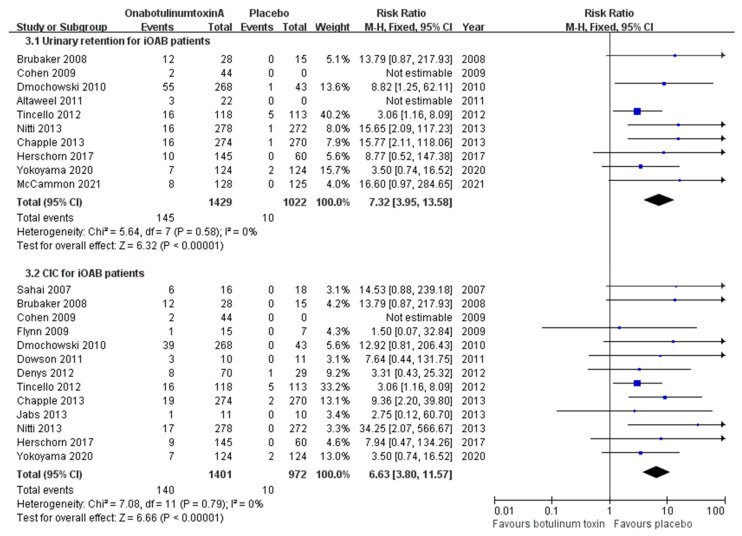
Forest plots of the incidence of urinary retention and clean intermittent catheterization in patients with idiopathic overactive bladder. CIC: clean intermittent catheterization; iOAB: idiopathic overactive bladder.

**Figure 5 toxins-16-00343-f005:**
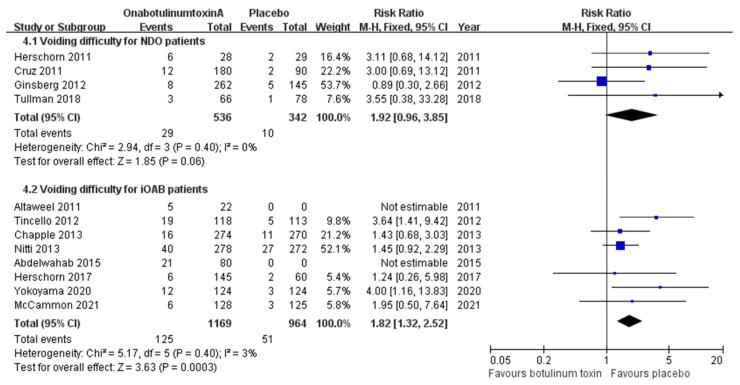
Forest plots of the incidence of voiding difficulty in patients with neurogenic detrusor overactivity and idiopathic overactive bladder. iOAB: idiopathic overactive bladder; NDO: neurogenic detrusor overactivity.

**Figure 6 toxins-16-00343-f006:**
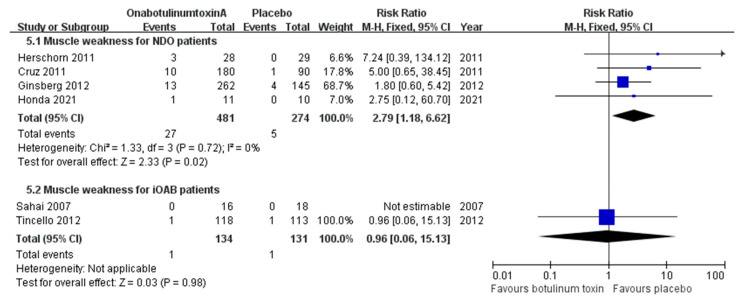
Forest plots of the incidence of muscle weakness in patients with neurogenic detrusor overactivity and idiopathic overactive bladder. iOAB: idiopathic overactive bladder; NDO: neurogenic detrusor overactivity.

**Table 1 toxins-16-00343-t001:** Characteristics of the individual studies.

First Author	Year	Region	No. of Patients (Female)	Ages, Means (SD)	Design	Classification	Basic Diseases	Randomization	AEs Follow-Up (Weeks)
Schurch [25]	2005	Switzerland, Belgium, and France	59 (23)	41	RCT	NDO	MS6, SCI53	BTX 200U 19, BTX 300U 19, Placebo 21	2, 6, 12, 18, 24
Herschorn [26]	2011	Canada	57 (23)	42.8	RCT	NDO	MS19, SCI38	BTX 300U 28, Placebo 29	6, 24, 36
Cruz [23]	2011	Europe, North America, Latin America, South Africa, and Asia Pacific	275 (155)	46 (13.1), 44.4 (13.9), 46.9 (13.4)	RCT	NDO	MS154, SCI121	BTX 200U 92, BTX 300U 91, Placebo 92	At least 12
Ginsberg [24]	2012	USA and Europe	416 (245)	46 (13)	RCT	NDO	MS227, SCI189	BTX 200U 135, BTX 300U 132, Placebo 149	At least 12
Ginsberg [27]	2013	Europe, North America, Latin America, South Africa, and Asia Pacific	691 (400)	45.9	RCT	NDO	MS381, SCI310	BTX 200U 227, BTX 300U 223, Placebo 241	At least 12
Rovner [28]	2013	Europe, North America, Latin America, South Africa, and Asia Pacific	691 (400)	45.9 (13.3), 45.6 (13.0), 46.2 (13.3)	RCT	NDO	MS381, SCI310	BTX 200U 227, BTX 300U 223, Placebo 241	At least 12
Tullman [29]	2018	Europe, North America	144 (127)	51.6 (10.3)	RCT	NDO	MS144	BTX 100U 66, Placebo 78	2, 6, 12
Honda [30]	2021	Japan	21 (4)	50.9 (14.1), 47.2 (18.3)	RCT	NDO	SCI21	BTX 200U 11, Placebo 10	At least 12
Sahai [31]	2007	UK	34 (19)	49.8, 50.8	RCT	iOAB	NA	BTX 200U 16, Placebo 18	4, 12, 24
Brubaker [32]	2008	USA	43 (43)	64.7 (14.5), 69.2 (13.5)	RCT	iOAB	NA	BTX 200U 28, Placebo 15	Within 52
Cohen [33]	2009	USA	44	NA	RCT	iOAB	NA	BTX 150U 22, BTX 100U 22	2, 6, 12, 24
Flynn [34]	2009	USA	22	66	RCT	iOAB	NA	BTX 200U or 300U 15, Placebo 7	3, 6
Dmochowski [35]	2010	North America, Europe	313 (288)	58.8	RCT	iOAB	NA	BTX 50U 57, BTX 100U 54, BTX 150U 49, BTX 200U 53, BTX 300U 56, Placebo 44	2, 6, 12, 18, 24, 30, 36
Altaweel [36]	2011	Saudi Arabia	22	NA	RCT	iOAB	NA	BTX 100U 11, BTX 200U 11	2, 12
Dowson [37]	2011	UK	23	NA	RCT	iOAB	NA	BTX 100U 10, Placebo 13	4, 12
Rovner [38]	2011	North America, Europe	313 (288)	58.8	RCT	iOAB	NA	BTX 50U 57, BTX 100U 54, BTX 150U 49, BTX 200U 53, BTX 300U 56, Placebo 44	2, 6, 12, 18, 24, 30, 36
Denys [39]	2012	France	99 (87)	61.6 (14.0)	RCT	iOAB	NA	BTX 50U 23, BTX 100U 23, BTX 150U 30, Placebo 31	4, 12, 20, 24
Tincello [40]	2012	UK	240 (240)	60.7, 58.2	RCT	iOAB	NA	BTX 200U 122, Placebo 118	6, 12, 24
Chapple [41]	2013	USA and Europe	548 (473)	59.5 (15.5), 59.2 (14.1)	RCT	iOAB	NA	BTX 100U 277, Placebo 271	Within 24
Nitti [42]	2013	USA, Canada	557 (497)	61.7 (12.7), 61 (13.1)	RCT	iOAB	NA	BTX 100U 280, Placebo 277	Within 24
Jabs [43]	2013	Canada	21 (21)	63 (9.4), 63.8 (11.2)	RCT	iOAB	NA	BTX 100U 11, Placebo 10	6, 12, 24
Sievert [44]	2014	Europe, North America	1105 (970)	60.6 (14.2), 60.1 (13.6)	RCT	iOAB	NA	BTX 100U 557, Placebo 548	Within 24
Abdelwahab [45]	2015	Egypt	80 (63)	30.22 (8.37), 31.35 (7.61)	RCT	iOAB	NA	BTX 100U 40, BTX 200U 40	4, 12, 24, 36
Herschorn [46]	2017	North America, Europe	356 (308)	62.0 (12.3)	RCT	iOAB	NA	BTX 100U 145, Solifenacin 151, Placebo 60	2, 6, 12, 18, 24
Yokoyama [47]	2020	Japan	248 (186)	65.6 (12.4), 66.2 (12.2)	RCT	iOAB	NA	BTX 100U 124, Placebo 124	12
McCammon [48]	2021	USA	254 (226)	60.8 (12.4)	RCT	iOAB	NA	BTX 100U 129, Placebo 125	12

AE: adverse effect; BTX: onabotulinumtoxinA; iOAB: idiopathic overactive bladder; MS: multiple sclerosis; NA: not applicable; NDO: neurogenic detrusor overactivity; RCT: randomized controlled trial; SCI: spinal cord injury; SD: standard deviation.

**Table 2 toxins-16-00343-t002:** Risks of bias in the individual studies.

First Author	Year	Random Sequence Generation (Selection Bias)	Allocation Concealment (Selection Bias)	Blinding of Participants and Personnel (Performance Bias)	Blinding of Outcome Assessment (Detection Bias)	Incomplete Outcome Data (Attrition Bias)	Selective Reporting (Reporting Bias)	Other Bias
Schurch	2005	Low	Low	Low	Low	Low	Low	Unclear
Herschorn	2011	Low	Low	Low	Low	Low	Low	Unclear
Cruz	2011	Low	Low	Low	Low	Low	Low	Unclear
Ginsberg	2012	Low	Low	Low	Low	Low	Low	Unclear
Ginsberg	2013	Low	Low	Low	Low	Low	Low	Unclear
Rovner	2013	Low	Low	Low	Low	Low	Low	Unclear
Tullman	2018	Low	Low	Low	Low	Low	Low	Unclear
Honda	2021	Low	Unclear	Low	Low	Low	Low	Unclear
Sahai	2007	Low	Low	Low	Low	Unclear	Low	Unclear
Brubaker	2008	Unclear	Unclear	Low	Unclear	Low	Low	Unclear
Cohen	2009	Low	Low	Low	Unclear	Low	Low	Unclear
Flynn	2009	Low	Low	Low	Unclear	Low	Low	Unclear
Dmochowski	2010	Low	Unclear	Low	Low	Low	Low	Unclear
Altaweel	2011	Low	Low	Low	Low	Low	Low	Unclear
Dowson	2011	Low	Low	Low	Low	Unclear	Low	Unclear
Rovner	2011	Low	Low	Unclear	Low	Low	Low	Unclear
Denys	2012	Low	Unclear	Low	Low	Low	Low	Unclear
Tincello	2012	Low	Low	Low	Low	Low	Low	Unclear
Chapple	2013	Low	Low	Low	Unclear	Low	Low	Unclear
Nitti	2013	Low	Low	Unclear	Low	Low	Low	Unclear
Jabs	2013	Unclear	Low	Low	Low	Low	Low	Unclear
Sievert	2014	Unclear	Low	Low	Unclear	Low	Low	Unclear
Abdelwahab	2015	Low	Unclear	Low	Unclear	Low	Low	Unclear
Herschorn	2017	Unclear	Unclear	Low	Low	Low	Low	Unclear
Yokoyama	2020	Unclear	Low	Low	Low	Low	Low	Unclear
McCammon	2021	Low	Low	Low	Low	Low	Low	Unclear

## Data Availability

The data supporting the conclusions of this article will be made available by the authors on request.

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
