# Peer review of "Adverse Effects of Intravesical OnabotulinumtoxinA Injection in Patients with Idiopathic Overactive Bladder or Neurogenic Detrusor Overactivity: A Systematic Review and Meta-Analysis of Randomized Controlled Studies"

_toxins, 2024, doi:10.3390/toxins16080343_

Round 1

Reviewer 1 Report

Comments and Suggestions for Authors

The review deals with a potentially important topic, as it presents the current knowledge on adverse effects that may appear due to intravesical injections of OnabotulinumtoxinA in patients with iOAB and NDO. Although some review articles concerning the efficiency and side effects of OnabotulinumtoxinA in urology have been published, none of them focus on potential adverse effects comprehensively. The paper is interesting, concise, and well-written. I have only some minor concerns:

1.      Abstract:

·         Line 14: most probably it should be “8 on NDO and 18 on iOAB”, not OAB

·         Line 15: please explain RR, as it is only explained in the Results section

2.      Results:

·         Page 3 of 22: I would suggest moving Figure 1 from the Results section to the Materials and Methos section, as while reading Results and analyzing the Figure it is not clear why some of the records were excluded (and it is all described in Materials and Methods).

·         Line 94: the punctuation mark is missing after Dmochowski et al. (most probably a dot?).

·         Table 1: although the abbreviation NA is commonly used and known, I would still recommend to explain it in the Table 1 description

3.      Discussion:

·         Lines 356-357: the part “…patients who had never undergone CIC previously requiring CIC” is unclear to me. Can You explain?

Author Response

We are grateful to you for deeming our work suitable for publication in Toxins. We have carefully studied your constructive comments and provide responses below.

Comment 1: Abstract line 14: most probably it should be “8 on NDO and 18 on iOAB”, not OAB
Response 1: Based on your correction, we have adjusted the text here.

Comment 2: Abstract line 15: please explain RR, as it is only explained in the Results section 
Response 2: We've added "relative risk" at the first occurrence of the term "RR".

Comment 3: Results page 3 of 22: I would suggest moving Figure 1 from the Results section to the Materials and Methods section, as while reading Results and analyzing the Figure it is not clear why some of the records were excluded (and it is all described in Materials and Methods).
Response 3: 
     The paragraph order in the journal "Toxins" differs from other journals, with the Results section placed before the Materials and Methods section. As a result, readers encounter the research results before understanding the background knowledge of the research methods.
     After discussion between us, the two authors, we decided to keep Figure 1 in its current position for two main reasons. Firstly, Figure 1 indicates the number of articles corresponding to each step of meta-analysis, and the number of articles in each step is presented in the Results section. Secondly, when we attempted to move Figure 1 to the Materials and Methods section, the layout was disrupted due to the need to switch the orientation of Table 1 and Table 2 pages to landscape. Thus, we ultimately decided to keep Figure 1 in its current position.

Comment 4: Results line 94: the punctuation mark is missing after Dmochowski et al. (most probably a dot?).
Response 4: Since the sentence ends with Dmochowski et al., the abbreviation period and the sentence period are used only once. However, the next sentence starts with Chapple et al., which indeed looks confusing, making it unclear for the reader where the sentence ends. We have added "in addition" to help separate the sentences and avoid confusion.

Comment 5: Table 1: although the abbreviation NA is commonly used and known, I would still recommend to explain it in the Table 1 description 
Response 5: Based on your suggestion, we have adjusted the description here.

Comment 6: Discussion line 356-357: the part “…patients who had never undergone CIC previously requiring CIC” is unclear to me. Can You explain?
Response 6: The meaning here is that if patients who were originally CIC-naive receive onabotulinumtoxinA treatment, there is a high likelihood that they will require CIC after onabotulinumtoxinA treatment.

Reviewer 2 Report

Comments and Suggestions for Authors

Thank you for the paper, Adverse Effects of Intravesical OnabotulinumtoxinA Injection for Patients with Idiopathic Overactive Bladder or Neurogenic Detrusor Overactivity: A Systematic Review and Meta-analysis of Randomized Controlled Studies. This is a needed review of BT-injection, on adverse events. It was a well-conducted META analysis with clear endpoints for conclusions about the risks of BT treatments. Thus. it is an important review and will add to the field of neurourology. It is well-written and I have no further concerns

Author Response

We greatly appreciate the reviewer for taking the time to review our article, and we are also thankful for your positive feedback.

Reviewer 3 Report

Comments and Suggestions for Authors

The authors present a systematic review on the adverse events related to intra-detrusor Botox injection for idiopathic overactive bladder or neurogenic detrusor overactivity. They found that minor adverse events were common and major adverse events were rare. While this is certainly not novel or surprising information for urologists or urogynecologists, it is helpful to have the expected rates of each adverse event reported. The study is well designed, and the manuscript is well written.

Author Response

We are grateful to the reviewer for taking your valuable time to review our article. We also deeply appreciate your positive evaluation of our work.